# Engineering topological states in atom-based semiconductor quantum dots

M. Kiczynski[1,2], S. K. Gorman[1,2], H. Geng[1,2], M. B. Donnelly[1,2], Y. Chung[1,2], Y. He[1,3], J. G. Keizer[1,2] & M. Y. Simmons[1,2 ✉]

The realization of controllable fermionic quantum systems via quantum simulation is instrumental for exploring many of the most intriguing effects in condensed-matter physics[1–3]. Semiconductor quantum dots are particularly promising for quantum simulation as they can be engineered to achieve strong quantum correlations. However, although simulation of the Fermi–Hubbard model[4] and Nagaoka ferromagnetism[5] have been reported before, the simplest one-dimensional model of strongly correlated topological matter, the many-body Su–Schrieffer–Heeger (SSH) model[6–11], has so far remained elusive—mostly owing to the challenge of precisely engineering long-range interactions between electrons to reproduce the chosen Hamiltonian. Here we show that for precision-placed atoms in silicon with strong Coulomb confinement, we can engineer a minimum of six all-epitaxial in-plane gates to tune the energy levels across a linear array of ten quantum dots to realize both the trivial and the topological phases of the many-body SSH model. The strong on-site energies (about 25 millielectronvolts) and the ability to engineer gates with subnanometre precision in a unique staggered design allow us to tune the ratio between intercell and intracell electron transport to observe clear signatures of a topological phase with two conductance peaks at quarter-filling, compared with the ten conductance peaks of the trivial phase. The demonstration of the SSH model in a fermionic system isomorphic to qubits showcases our highly controllable quantum system and its usefulness for future simulations of strongly interacting electrons.

Superconductivity, magnetism[12], low-dimensional electron transport[13], topological phases[14] and other exotic phases of matter arise owing to the presence of strongly interacting particles within crystals[15]. However, the complexity of simulating such large quantum systems becomes intractable using classical computing methods[16]. A promising solution is to build a physical system at the same scale so that we can simulate these interacting fermionic systems[17,18] directly, known as analogue quantum simulation[19,20]. The Su–Schrieffer–Heeger (SSH) model is the prototypical example of topological matter that describes a single electron hopping along a one-dimensional dimerized lattice with staggered tunnel couplings, $v$ and $w$, as shown in Fig. 1a[21]. The SSH model has been experimentally simulated in physical systems of varying dimensions from Rydberg atoms (about 10 μm) to mechanical systems (about 10 mm) (Table 1). The coupling strengths of the various simulators lie in the nanoelectronvolt to microelectronvolt range, limiting their ability to reach the fully coherent regime. Importantly, these systems can be readily solved classically as they do not simulate many-body interactions. Only recently has the interacting many-body SSH model been observed using Rydberg atoms with an effective infinite on-site interaction (hardcore bosons)[10]. The ability to control the interaction strength, however, is critical for investigating fermionic systems[22,23].

Semiconductor quantum dots are an emerging platform for the quantum simulation of strongly correlated electron systems[4,5,20]. Interacting electrons confined to quantum dots have been described by the Hubbard model[16] involving Coulombic interactions that describe the energy required to add electrons to the same (on-site, $U$) or neighbouring (intersite, $V$) quantum dot[24]. Here intersite hopping is governed by the tunnel coupling, $t$, between quantum dots, and each dot can be tuned using electrostatic gates to raise or lower their energy levels, $\epsilon$ (ref. [4]). Phosphorus donors in silicon in particular have been proposed as promising candidates for simulators as they are nanoscale in size with very strong on-site energies ($U \approx 25$ meV) and can be engineered to have strong intersite ($V \approx 5$ meV) and hopping ($t \approx 5$ meV) energies, while operating with a low thermal energy of $k_B T \approx 0.02$ meV (ref. [25]), where $k_B$ is the Boltzmann constant and $T$ is the temperature, reaching a range of $U/t \approx 1$–100 while remaining in the low-temperature limit with $t/k_B T > 10$ (ref. [23]). The ability to reach the low-temperature, strongly interacting regime allows for a number of coveted quantum phases, such as superconductivity[26] and antiferromagnetism[27], to be simulated[28]. Despite the promise of semiconductor simulators, significant challenges have remained to simulate full quantum systems. These relate to the ability to precisely engineer, and tune, both the large on-site interaction energies and tunnel couplings to allow for the

[1]Centre of Excellence for Quantum Computation and Communication Technology, School of Physics, UNSW Sydney, Kensington, New South Wales, Australia. [2]Silicon Quantum Computing Pty Ltd, UNSW Sydney, Kensington, New South Wales, Australia. [3]Present address: Shenzhen Institute for Quantum Science and Engineering, Southern University of Science and Technology, Shenzhen, China. ✉e-mail: michelle.simmons@unsw.edu.au

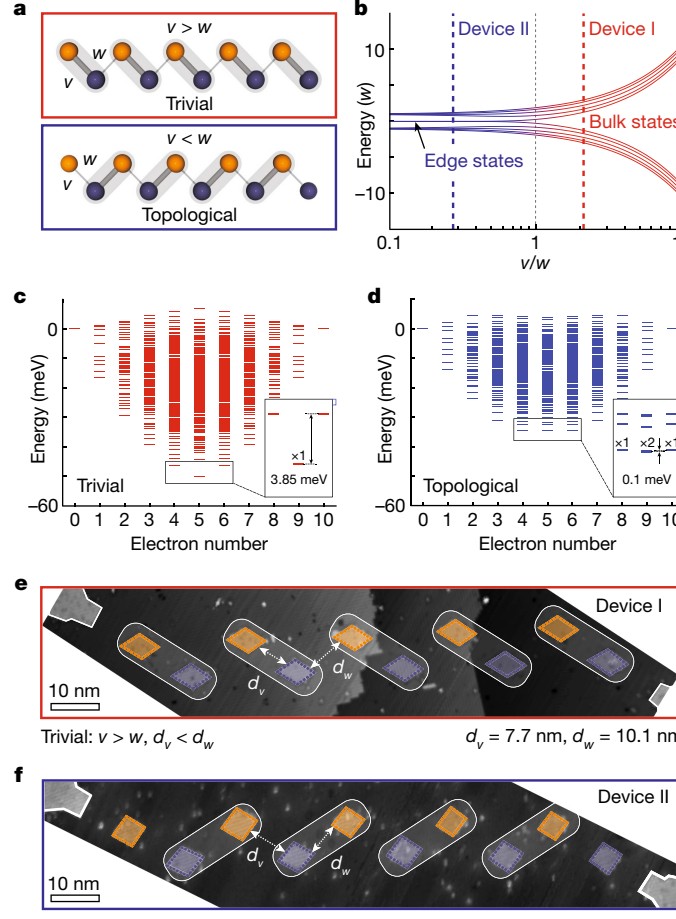

**Fig. 1 | Realization of the SSH model in precision-engineered phosphorus donors in silicon. a**, A dimerized one-dimensional lattice with staggered hopping amplitudes (tunnel couplings) $v$ and $w$. The array consists of two sublattices where there exists a trivial phase, $v > w$ (top) and a topological phase $v < w$ (bottom). **b**, Single-particle energy spectrum of the SSH model for a linear array of ten quantum dots as a function of the interdot coupling ratio $v/w$. For $v/w < 1$, there exists zero-energy topological edge states, whereas the trivial case $v/w > 1$ exhibits an excitation gap. **c**, **d**, Calculated multi-electron energy spectrum in the trivial (**c**) and topological (**d**) phases for different electron numbers at quarter-filling. The trivial array exhibits a single ground state with five electrons about 3.85 meV below the four- and six-electron states, whereas the topological phase exhibits a nearly four-fold degeneracy involving four, five (two-fold degenerate) and six electrons. **e**, **f**, Scanning tunnelling micrograph for the trivial (**e**; device I) and topological (**f**; device II) phases. The lighter regions show the open lithographic hydrogen mask. The devices consist of an array of $N = 10$ Coulomb-confined quantum dots with staggered nearest-neighbour distances, tunnel-coupled to a source (drain) lead at the start (end) of the array outlined in white to perform bias spectroscopy. Device I is designed to be in the trivial phase with $d_v = 7.7 \pm 0.1$ nm and $d_w = 10.1 \pm 0.2$ nm, and device II is designed to be in the topological phase with $d_v = 9.6 \pm 0.4$ nm and $d_w = 7.8 \pm 0.6$ nm.

formation of a well defined coherent state across the system. In particular, for 10 quantum dots, we require precision control across 110 different experimental parameters related to $U$, $V$, $t$ and $\epsilon$.

In this paper, we utilize the atomic-precision placement accuracy of the scanning tunnelling microscope (STM) to engineer quantum dots with large on-site energies ($U \approx 25$ meV) and uniform size to realize a homogeneous linear array for reliable simulation accuracy. If the quantum dots are too big, the capacitive coupling between individual dots becomes too large to independently control them. Conversely, if the they are too small, then a small change in the number of

phosphorus donors within the quantum dot can substantially change the on-site energy, leading to randomness in the array. Importantly, our subnanometre-precision capability allows us to change the values of $v$ and $w$ with millielectronvolt resolution so that we can reliably enter both topologically trivial and topologically non-trivial regimes. Finally, a substantial challenge for gate-defined quantum dot architectures is that they require electrostatic gates to create the quantum dot potential and control the tunnel couplings requiring a minimum of about two gates per quantum dot[29–32]. With donor-based dots, we do not require these additional confinement gates and require only six electrostatic gates to control a ten-quantum dot array, thereby avoiding unnecessary cross-talk between gates. To ensure the creation of a well defined quantum state across the array, we designed an iterative maximum-current-alignment procedure to align the quantum dot energy levels within approximately 0.5 meV. The quantum state formed is then measured using bias spectroscopy via the planar source and drain leads. Having determined the necessary conditions to form the desired state, we simulate the one-dimensional topological phases associated with the interacting SSH model[21].

The SSH model is one of the simplest known instances of topological quantum systems. The eigenenergies (Fig. 1b) of the SSH model give rise to two distinct phases dependent on the ratio of tunnel couplings with a topological phase transition at $v = w$. For $v > w$, there is a topologically trivial phase, where the lattice acts as a bulk insulator with the electron delocalized across the array and an energy gap between the upper and lower bulk states. For $v < w$, there exists a topologically non-trivial phase—a symmetry-protected topological phase—which gives rise to two zero-energy edge states where the electron is localized at the edge sites of the lattice[33].

In previously measured instances of the SSH model, particle–hole symmetry was conserved owing to a lack of intersite electron–electron interactions[21]. However, quantum dots in semiconductors are affected by the intersite Coulomb interaction, $V_{i,j}$ which is the change in energy of quantum dot $j$ owing to the addition of an electron on quantum dot $i$. These long-range electron–electron interactions break the particle–hole symmetry, leading to non-degenerate electron and hole states[4]. As a consequence, it is important to control both the electron filling of the array and changes in the electrostatic environment to ensure correct simulation results. This requires us not only to have independent control of each quantum dot potential but also to alter the energy levels in unison—a tremendous technological challenge for such a small array with strong tunnel couplings and on-site energies (Supplementary Section II). As a result, we consider the full Hamiltonian of the extended (spinless) Hubbard model for a linear array of $N$ quantum dots, given by

$$H_U = \sum_{i=1}^{N} \epsilon_i n_i + \sum_{i}^{N} U_i n_i (n_i - 1) + \sum_{i}^{N-1} t_{i,i+1}(c_i^\dagger c_{i+1} + \text{h.c.}) + \sum_{i,j}^{N} V_{i,j} n_i n_j, \quad (1)$$

where $\epsilon_i$ is the energy levels of the $i$th dot of the array, $n_i$ is the dot occupation operator, $t_{i,i+1}$ is the tunnel coupling between nearest-neighbour $i$th and $i$th + 1 dots, $U_i$ is the on-site Coulomb interaction term, $V_{i,j}$ is the intersite Coulomb interaction terms between the $i$th and $j$th dots, $c_i^\dagger$ ($c_i$) are the creation (annihilation) operator of a particle on site $i$ and h.c. indicates Hermitian conjugate.

In Fig. 1c, d, we show the 1,024 calculated multi-electron energy levels based on the two experimental arrays fabricated in the trivial and topological phases respectively, ordered by the number of electrons in the quantum dot array at quarter-filling. For ten quantum dots, in which each dot can host two electrons, quarter-filling corresponds to where there are five electrons shared across the dots. Here, if the array is in the trivial phase, the electrons will simply arrange themselves into each dimer such that they are spread evenly across the array. In contrast, if the array is in the topological configuration at quarter-filling, four electrons arrange themselves in the middle of the array (similar to the trivial phase). However, the fifth electron cannot occupy a dimer.

## Table 1 | Experimental demonstrations of the SSH model to date

| Physical system | Physical size | Physical separation | Mean $t$ (meV) | $v/w$ | $U$ (meV) | $V$ (meV) | Sites ($N$) |
|---|---|---|---|---|---|---|---|
| Mechanical[9] | 18 mm | N/A | $2\times10^{-8}$ | ~0.8–1.2 | – | – | 27 |
| Superconductor[11] | 500 μm | ~20 μm | $1\times10^{-5}$ | 0.20–5.0 | – | – | 5 |
| LC waveguides[39] | 290 μm | 2–10 μm | $1.5\times10^{-3}$ | 0.56 | – | – | 9 |
| Rydberg atoms[7] | ~80 μm BEC | N/A | $4\times10^{-9}$ | 0.11–0.90 | – | – | 20 |
| Nanomechanical[40] | 3 μm | 0.5 μm | $2\times10^{-10}$ | 0.25–4.0 | – | – | 8 |
| Micropillars[8] | ~2 μm | ~1 μm | 1.7 | 0.15 and 6.7 | – | – | 20 |
| Hardcore bosons[10] | ~1 μm | 8–12 μm | $7\times10^{-6}$ | 0.38 | ∞ | – | 14 |
| Nanoparticles[41] | 540 nm | <50 nm | NA | 0.33 | – | – | 7 |
| Si:P (this work) | 5 nm | 7–11 nm | 3.4 | 0.265 and 2.08 | ~25 | <5 | 10 |

Physical size and separation, mean intersite couplings ($t$) and their staggered ratio ($v/w$), on-site ($U$) and intersite ($V$) interaction energies, and number of modelled sites, $N$. BEC, Bose–Einstein condensate; LC, inductance–capacitance; N/A, not available.

We focus on quarter-filling of the array as the interacting topological states involve this fifth electron becoming localized to each end of the array. In the trivial phase ($v > w$), there is a singly degenerate ground state (box in Fig. 1c) delocalized across the entire array with a large energy gap of about 3.85 meV separating the five-electron state from the four- and six-electron states. The topological phase ($v < w$) exhibits a nearly four-fold degenerate (about 0.1 meV) ground state (box in Fig. 1d) involving four, five (two-fold degenerate) and six electrons. The ground state corresponds to four electrons on the inner eight quantum dots of the array with either zero, one or two additional electrons localized at the two edge quantum dots.

The experimental realization of two such arrays is shown in Fig. 1e, f, where the quantum dots are tunnel-coupled to source and drain leads to perform bias spectroscopy through the array. The regions outlined in orange and blue are the quantum dots, and the two regions at the start and end of the array outlined in white are the source and drain leads. The tunnel couplings $t_{i,i+1}$ are engineered via the interdot separation, $d_{i,i+1}$ and follow an inverse exponential dependence, $t_{i,i+1} \propto \exp(-2d_{i,i+1}/3)$ (ref. [34]). By staggering the quantum dot array, we have ensured that non-nearest-neighbour tunnelling is exponentially suppressed with an estimated $t_{i,i+2}/t_{i,i+1} \approx 0.01$, ensuring that electron transport occurs in series through the array, while maximizing the differential lever arms to the dots (Supplementary Section I). The SSH model requires that the tunnel couplings have alternating strengths to observe the different topological phases while being simultaneously large enough to allow for a measurable transport current for bias spectroscopy. The quantum dot size is critical as the confinement potential experienced by the last outer electron, and hence the wavefunction overlap to the neighbouring quantum dots, depends on the number of donors comprising the quantum dot. We therefore fabricate quantum dots with an area of about 25 nm² (about 25 phosphorus donors per site[35]). The nanoscale size of the quantum dots allows us to achieve large on-site energies of about 25 meV, but, importantly, where a small change in the size of the quantum dot does not significantly alter $U$, $V$, or $t$. The small separation of ≤10 nm also allows us to achieve large tunnel couplings $t \approx 1$–10 meV for these quantum dot sizes[35]. Device I (Fig. 1e) is designed to be in the trivial phase with average staggered quantum dot distances, $d_v = 7.7 \pm 0.1$ nm and $d_w = 10.1 \pm 0.2$ nm, corresponding to $\langle v/w \rangle = 2.08$. Device II (Fig. 1f) is designed to be in the topological phase where we now engineer an average staggered donor distance, $d_v = 9.6 \pm 0.4$ nm and $d_w = 7.8 \pm 0.6$ nm, corresponding to $\langle v/w \rangle = 0.265$. These values highlight the subnanoscale accuracy that we can engineer devices with using STM lithography so that we can change $\langle v/w \rangle$ between 0.265 and 2.08 (ref. [36]).

Figure 2a shows an STM image of the full device I. Here the outlined lighter regions show the lithographic hydrogen mask with six capacitively coupled control gates (G1 to G6), crucial to independently control the energy levels of the quantum dots (device II gate structure is nominally identical). Owing to the unique geometry of the device, the total lever arms of all gates linked together to each quantum dot are engineered to be consistent with a variation of less than 2.5%. This small variation means that we can also raise the global energy level of the whole quantum dot array for bias spectroscopy to measure the different phases of the SSH model. To align the energy levels of the quantum dots, we used a maximum-current-alignment scheme, in which the individual gates are tuned as outlined in Fig. 2b. This is achieved by initially setting the gate voltages at a conductance peak determined by sweeping G1, G2 and G3, against G4, G5 and G6, while measuring the current through the array. While positioned at this conductance peak, each gate is then individually swept around a set value, whereas all other gates were kept constant, as illustrated in Fig. 2c. After sweeping all six gates about their set voltage, the largest current peak is found and the corresponding gate is updated to the voltage at the centre of the current peak (G5 in the first iteration shown in Fig. 2d). All gates are then swept again, repeating this process, updating a single gate at a time as shown in Fig. 2d. Figure 2e shows the maximum current measured on each gate sweep per iteration for a constant source–drain bias, $V_{SD}$. When the maximum current plateaus, $V_{SD}$ is reduced further and the entire process is repeated to increase the alignment accuracy.

Once the energy levels of the quantum dots are aligned, we perform a stability diagram measurement by shifting the energy levels of all quantum dots to investigate the electron occupation of the array. The stability diagram allows us to determine the electron occupation of the array as a function of the energy levels of the quantum dots. At zero source–drain bias (dashed white line in Fig. 2b), there is only enough energy to add a single electron to the array at a time. In this regime, we simulate the SSH model in the two devices. Figure 3 shows a comparison of the experimental and theoretical results of the two devices. Figure 3a shows the calculated normalized zero-bias conductance as a function of the ratio of tunnel couplings ($v/w$) of the array. The design of device I in the trivial phase is given by the dashed red line ($\langle v/w \rangle = 2.08$), whereas device II in the topological phase is given by the dashed blue line ($\langle v/w \rangle = 0.265$).

Figure 3b shows the zero-bias conductance as a function of the combined voltages on all the gates obtained from the trivial phase (device I) with the theoretical calculation shown in red. There are ten conductance peaks corresponding to a change in the total number of electrons on the array (Fig. 3c). We control the electron filling of the array by adjusting the gate voltages to tune the electron number from $m$ to $m + 10$ (half-filling). At a quarter-filling ($m + 5$), there is a gap in the energy spectrum (7.9 meV, approximately twice the gap of 3.85 meV in Fig. 1c) corresponding to the single ground state of the SSH model for the trivial phase (Fig. 1c). From the estimate of $V_{i,j}$ from electrostatic modelling, and fitting the magnitude of the tunnel coupling (Methods), we model the array to obtain the width of the different electron number regions, $S_k$ (width of the $m + k \to m + k + 1$ region). Figure 3d shows the

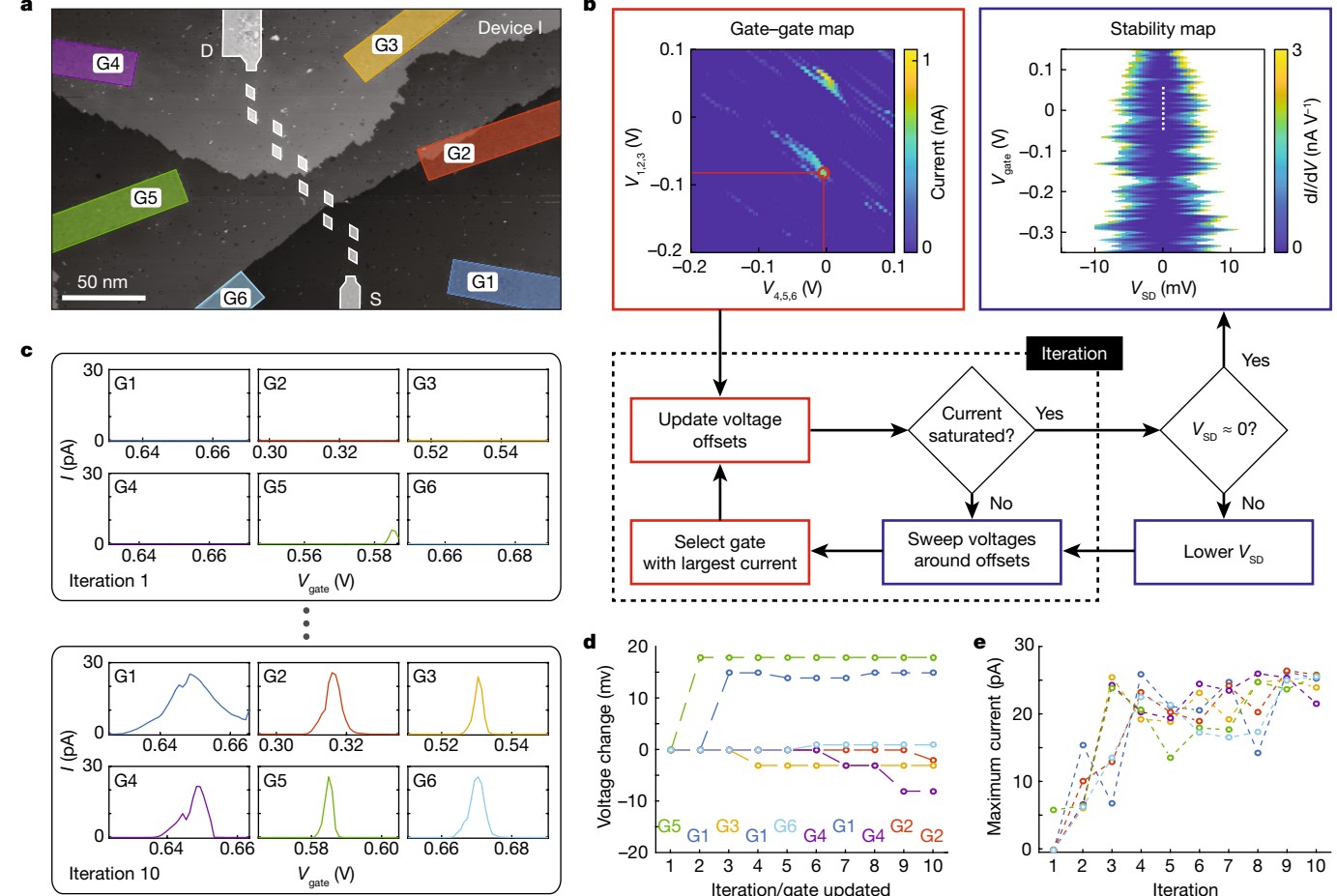

**Fig. 2 | Maximum-current-alignment scheme of the quantum dot array using only six in-plane gates. a**, An STM micrograph of device I, which shows the six control gates, labelled G1 to G6, used to tune the energy levels of the quantum dots to observe the conductance peaks using bias spectroscopy. Here we can observe the atomic step height of the silicon surface and the nanoscale size of the array. **b**, Schematic of the protocol used to align the quantum dots in the array. The quantum dots can be brought into alignment by varying the voltages applied to the control gates to tune the quantum dots for maximum current through the array. Each control gate is initially set to a specific value, chosen from a conductive region from the current map (red circle) while changing gates G1–G3 and G4–G6 together. The voltage on each gate is then swept, in turn, around their respective maximum current values, while the other gates are kept constant. After all gate voltages have been swept, a single gate value is then updated corresponding to the maximum current measured. The process is then repeated updating one gate each time. When the maximum current saturates, the source–drain bias, $V_{SD}$, is then reduced and the control gates are retuned again to increase the maximum current. Once the $V_{SD}$ is near zero, a stability diagram is measured as shown in the top right and the zero-bias conductance (dotted white line) is used for comparison with the simulated SSH model in Fig. 3. **c**, Examples of the individual gate sweeps on the first iteration (top) and on the tenth iteration (bottom) for a constant $V_{SD}$. **d**, **e**, The voltage on each gate per iteration (the gate updated per iteration is labelled at the bottom; **d**) and showing the maximum current measured on each gate sweep per iteration (**e**).

width of the experimentally measured stability regions, obtained by determining the energy between adjacent conductance peaks, compared with the theoretical calculations based on electrostatic modelling with a tunnel coupling ratio $\langle v/w \rangle = 2.08$. We find excellent agreement between the experimental and theoretical values. Small discrepancies ($\lesssim 1$ meV) are most likely due to small misalignments of the quantum dot energy levels, which gives rise to on-site disorder, causing small shifts in the conductance peaks such that the peak structure is no longer symmetric around zero (Supplementary Section III).

We now look at the topological phase of the SSH model (device II) shown in Fig. 3e. The blue line in Fig. 3e represents the theoretical fit to the experiment, with a tunnel coupling ratio $\langle v/w \rangle = 0.265$. We show a similar voltage range scan as for device I but here we observe only two sets of closely spaced peaks at zero gate voltage and at 85.5 mV corresponding to the average on-site energies across the array, $\langle U \rangle = 22.0 \pm 3.2$ meV. The conductance peaks from the states away from quarter-filling are not visible as they are now delocalized within the bulk of the array with a low probability of existing at the edge quantum

dots. As a result, tunnelling between these bulk-like states and the source and drain leads is significantly suppressed. In the topological phase, the quarter-filling gap almost disappears completely with a sharp transition from the $m + 4$ to $m + 6$ states given by only two conductance peaks separated by about 0.2 meV, shown in Fig. 3f. These electron states correspond to where there are no electrons ($m + 4$ electrons) on the edge quantum dots to where both dots are occupied ($m + 6$ electrons). Importantly at exactly quarter-filling ($m + 5$ electrons), there is a non-zero probability that either four ($P \approx 0.05$), five ($P \approx 0.30$ or $P \approx 0.60$ owing to the two-fold degeneracy) or six ($P \approx 0.05$) electrons exist on the array at the same time owing to the nearly four-fold-degenerate ground state, as illustrated in Fig. 3g. This remarkable observation, that at zero gate bias there is a superposition of the number of electrons on the edge quantum dots, is a result of the near-zero energy of the topological states of the array and is a distinctive property of the many-body SSH model. As these topological states are localized at the edge quantum dots, the current flowing through the array corresponds to an electron moving from one side of the array

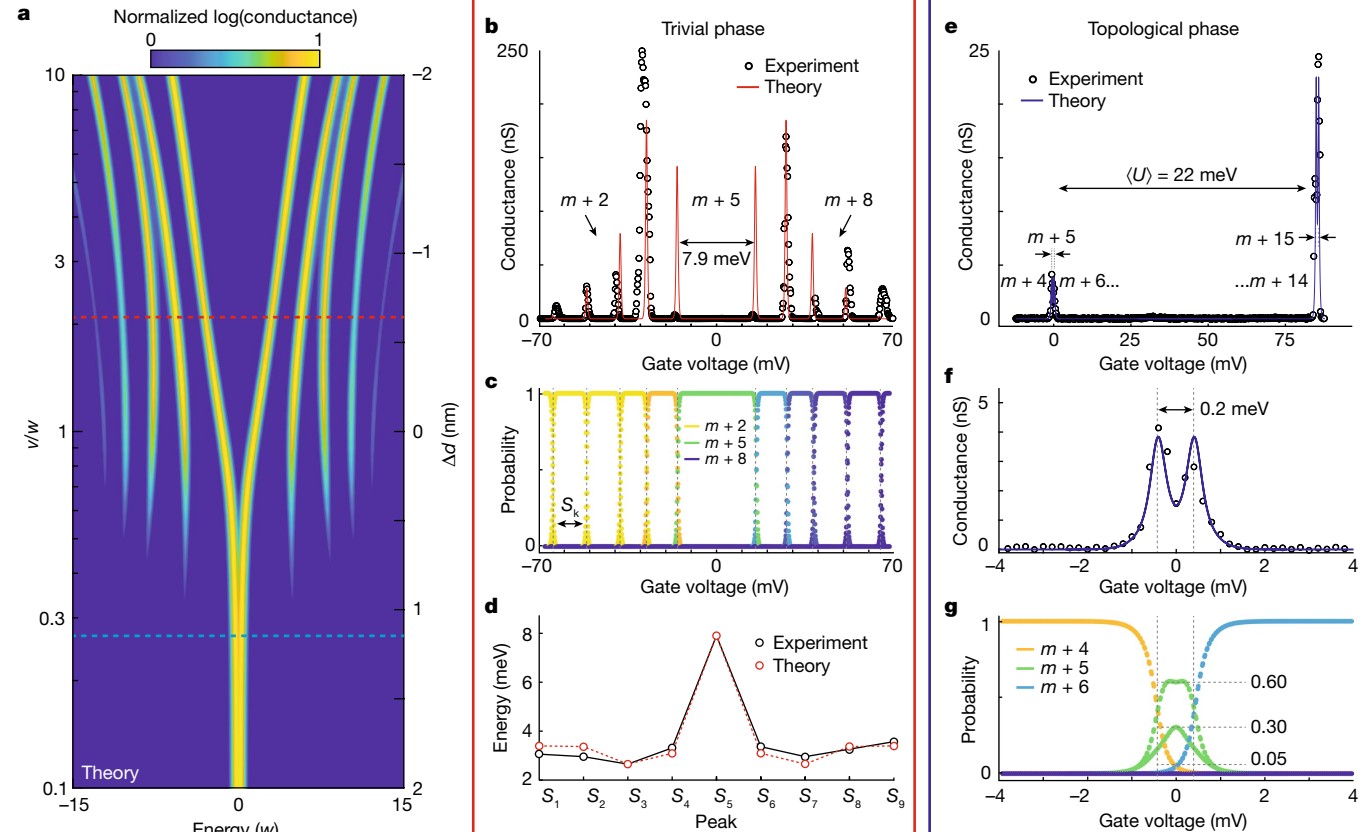

**Fig. 3 | Experimental signature of the SSH model in precision-engineered quantum dot arrays. a**, A theoretical map of the normalized log(conductance) as a function of the ratio of tunnel couplings (with the intersite Coulomb interactions given by a $1/d^{1.5}$ dependence, where $d$ is the quantum dot separation), with device I given by the dashed red line and device II given by the dashed blue line. **b**, Conductance trace obtained at zero source–drain bias ($V_{SD} = 0$), while shifting the energy levels of all quantum dots, in the trivial phase. We observe 10 conductance peaks corresponding to a single Hubbard band with a gap of about 7.9 meV (about $2 \times 3.85$ meV) at quarter-filling. **c**, The occupation probability of the many-body eigenenergies (all 1,024) of the Hubbard model as a function of the combined gate voltage for the trivial phase. The conductance peaks in **b** correspond to transitions between the different electron number ground states (grey dashed lines), which are separated in gate voltage by $S_k$. **d**, We use the extracted values of $S_k$ from the experimental results to obtain the parameters from the Hubbard model (Methods) and compare them to theory. **e**, In the topological phase, we observe two close conductance peaks around zero gate voltage resulting from the topological edge states at

quarter-filling, whereas no conductance peaks are observed away from quarter-filling (between 0 mV and 85 mV on the gates). The conductance peaks (there are two closely spaced) observed around 85 mV correspond to the addition of 10 extra electrons to the array and are separated by $\langle U \rangle$, which is the average on-site energies of the array. **f**, A zoom-in of the two conductance peaks of the topological phase corresponding to the $m + 4 \rightarrow m + 5$ and $m + 5 \rightarrow m + 6$ electron transitions to the array shown at 0 mV in **e**. **g**, The occupation probability of the topological phase as a function of the combined gate voltage. There is a sharp transition from $m + 4$ electrons where no electrons exist on the two edge sites of the array to $m + 6$ electrons in which the edge sites of the array are fully occupied. At quarter-filling ($m + 5$ electrons), there is a non-zero probability that there is either 4 ($P \approx 0.05$), 5 ($P \approx 0.30$ and $P \approx 0.60$ owing to the two-fold degeneracy) or 6 ($P \approx 0.05$) electrons existing on the array as a result of the nearly four-fold degenerate ground state of the topological phase. This remarkable feature of the two conductance peaks separated by about 0.2 meV is the signature of the topological phase of the many-body SSH model.

to the other without occupying any of the inner quantum dots. This unique property is a direct consequence of the topology embedded within the SSH model as confirmed by the double conductance peak in Fig. 3f.

In conclusion, we have observed clear signatures of the topological states of the interacting SSH model in semiconductor quantum dots. To achieve this, we have precision engineered two devices with sub-nanometre resolution consisting of a linear array of ten donor-based quantum dots, with staggered nearest-neighbour tunnel couplings. The minimal gate design of our epitaxial devices allows for both individual and global alignment of all the energy levels of the quantum dots such that bias spectroscopy can probe the topological phases of the array. Importantly, we confirm the existence of the one-dimensional topological phase at quarter-filling ($\langle v/w \rangle = 0.265$) with the observation of two overlapping peaks in the zero-bias conductance corresponding to the near four-fold degeneracy of the many-body SSH model. For the trivial phase ($\langle v/w \rangle = 2.08$), we observe ten zero-bias

conductance peaks corresponding to delocalized states across the entire array with an energy gap around quarter-filling owing to the large interdot coupling of about 3.5 meV. The low-gate-density design and low noise[37] exhibited in these all-epitaxial devices offers a promising platform for simulating strongly interacting electron systems for quantum chemistry applications[19]. Future work will focus on extending the size of the quantum dot arrays, incorporating charge sensors and extending the simulations to engineered two-dimensional lattices[38].

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

## Methods

### Device fabrication

The devices were fabricated on a 1–10 Ω cm p-type natural silicon substrate. The substrate was prepared via a series of high-temperature anneals, up to 1,100 °C, followed by a controlled cooling to 330 °C, resulting in a (2 × 1) reconstructed surface. The substrate was then terminated with atomic hydrogen, which can be selectively desorbed using the STM tip, leaving a hydrogen lithographic mask representing the device. After STM lithography, the substrate was exposed to a phosphine ($PH_3$) precursor gas, in which phosphorus was absorbed and incorporated, at 350 °C, into the exposed areas. The device was then encapsulated with 40 nm of natural silicon using molecular beam epitaxy at a rate of about 0.125 nm min$^{-1}$. A more detailed description of STM hydrogen lithography and the further device processing to electrically contact the device can be found in refs. [42–44].

### Experimental measurement setup

The electrical measurements were performed at millkelvin temperatures inside a dilution refrigerator. The two devices were measured in different fridges. Device I was measured in a fridge with a base temperature of about 10 mK, and device II was measured in a fridge with a base temperature of about 100 mK. Extended Data Fig. 1 shows a schematic of the electrical connections to the device. To perform the measurements, voltage sources were connected to the source and gates, to control their chemical potentials, and the transport current was measured through the drain. The drain current was amplified and converted to a voltage signal using a FEMTO variable-gain low-noise current amplifier DLPCA-200 with a low-pass 10-Hz filter. The filter signal was then digitized using a National Instrument data acquisition box (NIDAQ). The voltage sources used for the two devices were different. For device I, the source voltage was generated by the auxiliary of a Stanford Research Systems SR830 DSP lock-in amplifier and 1/50 room-temperature resistive-voltage divider, whereas the gate voltages were generated by two NIDAQs. Device II was measured on a different dilution refrigerator and had a different experimental setup. The source voltage was generated by a NIDAQ and a 1/50 room-temperature resistive-voltage divider. Gates 1, 2 and 3 voltages were also generated by the NIDAQ, whereas gates 4, 5 and 6 voltages were each generated by a Yokogawa 7651 programmable d.c. source. The different voltage sources used to measure the different devices should not affect the measurement results in a fundamental way.

### Calculation of the parameters of the extended Hubbard model

The quantum dot array can be described by the extended Hubbard model with the Hamiltonian given in equation (1). To theoretically solve the extended Hubbard model and calculate the parameters of the array, the data were fit using an open-source python package QmeQ[45]. The quantum dot Hamiltonian is described by the single-particle states $\sum_{i=1}^{N} \epsilon_i n_i + \sum_{i}^{N-1} t_{i,i+1}(c_i^{\dagger} c_{i+1} + \text{h.c.})$ and the Coulomb matrix elements, $\sum_{i,j}^{N} V_{i,j} n_i n_j$. The Hamiltonian is constructed in the Fock basis[45,46], for example, $|0010101110\rangle$, where 1 indicates an electron at that quantum dot and 0 indicates the quantum dot is unoccupied, and diagonalized exactly to obtain the exact many-body eigenstates $|a\rangle$, $H = \sum E_a |a\rangle\langle a|$. The transport current through the quantum dot array is then calculated numerically using the Pauli master equation. The array is assumed to be weakly coupled to the source and drain leads at a temperature, $T$, with a density of states that follows a Fermi distribution, $f(E)$.

Analogous to the non-interacting energy-level diagram shown in Fig. 1b, we also calculated the energy-level diagram of the interacting system. As with Fig. 3a, we assume that the intersite Coulomb terms follow a $d^{-1.5}$ dependence[47], and that the energy levels of the quantum dots are tuned via $\epsilon_i = -\sum_j^N V_{i,j}$. Extended Data Fig. 2 shows the energy spectrum for an array of ten sites for the $m + 4$, $m + 5$ and $m + 6$ electron states at quarter-filling, showing the more complex excited-state spectrum of the many-body states of the array.

For $v/w > 1$, at quarter-filling ($m + 5$ electrons), there is a large gap between the ground $m + 5$ and the lowest $m + 4$ and $m + 6$ electron states with a single ground state (labelled 'bulk states' analogous to Fig. 1b). By changing the tunnel couplings such that $v/w < 1$, the ground state becomes doubly degenerate with $m + 5$ electrons (labelled 'edge states' analogous to Fig. 1b). Here the quarter-filling gap is greatly reduced resulting in nearly degenerate states with differing electron numbers. The small energy gap at $v/w < 1$, is observed in the conductance trace in Fig. 3f and reflects the intersite Coulomb interactions present in the system.

In Fig. 3b and Fig. 3e, we show the theoretical calculation of the conductance through the quantum dot array fitted to the experimental data for the trivial and topological phases, respectively. In the theoretical calculation, we consider a spinless ten-dot array in the regime where there can be at most one electron on each of the quantum dots. Electron transport is restricted to sequential tunnelling through the array, as a result of the engineering of the device design, and only a single electron can tunnel through the quantum dots. No higher-order co-tunnelling events are allowed, as the source–drain bias is sufficiently small, making the choice of the Pauli master equation valid. All intersite Coulomb interaction terms, $V_{i,j}$, are included and accounted for by tuning the energy levels of the dots such that $\epsilon_i = -\sum_j^N V_{i,j}$.

The tunnel coupling and intersite Coulomb interaction parameters of the extended Hubbard model are obtained by fitting to the measured conductance peaks in the trivial phase, and using the measured distances from the STM and the electrostatic modelling. The trivial phase was used for the fit, as in this phase all ten conductance peaks can be observed, whereas for the topological phase only two conductance peaks are observed. The conductance peaks in the trivial phase in Fig. 3b each correspond to transitions between different particle number ground states, which are separated in gate voltage by $S_k$ (the voltage separation between the $k$ and $k + 1$ particle number). These peak separations, $S_k$, are dominated by the tunnel coupling strengths, $v$ and $w$, and the intersite Coulomb interaction strengths, $V_{i,j}$. Using the extracted values of $S_k$ from the experimental results, we fit the overall magnitude of the tunnel coupling and intersite Coulomb interactions (while keeping the theoretically determined trends as a function of distance) to find the parameters shown in Supplementary Tables 4–6.

The tunnel coupling, $t$, is engineered via the interdot donor separation, $d$, and follows an exponential dependence, $t = t_M \exp(-2d/3)$ eV, where the tunnel coupling magnitude $t_M = 0.1742$ for a 1P–2P quantum dot system[34]. We anticipate a similar dependence of $t$ as a function of interdot separation distance; however, $t_M$ is now used as a fitting parameter as it is known to depend on the crystallographic orientation of the quantum dots with $v$ (about $\langle 100 \rangle$) and $w$ (about $\langle 120 \rangle$) having different scaling factors owing to their different interdot axes. Supplementary Table 6 shows the distances and tunnel couplings for the two devices, with $\langle v/w \rangle = 2.08$ (1.702, 2.460) for the trivial phase and $\langle v/w \rangle = 0.265$ (0.146, 0.539) for the topological phase. The best fit to the zero-bias conductance peaks are achieved when the $w$ tunnel couplings are approximately twice the $v$ couplings for the same distance ($t_{M,w} \approx 2t_{M,v}$). This angular dependence on the tunnel coupling between individual donors is a well known consequence of the silicon crystal lattice[48–50]; however, it has now been observed directly for quantum dots of larger size. This additional knowledge will assist in future experimental designs for finer engineering of the tunnel couplings.

The extracted tunnel-coupling parameters give excellent fits to the experimental data as seen in Fig. 3b, e, with Fig. 3d showing a comparison of the peak separations, $S_k$, for the experimental and theoretical data. Small variations between the experimental and theoretical data can be attributed to errors in the electrostatic modelling and small offsets in the alignments of the quantum dots.

The presence of intersite Coulomb interactions gives rise to an approximately 0.2 meV splitting observed in the topological states as

seen in Fig. 3f. Extended Data Fig. 3 shows the calculated conductance for the topological phase for varying intersite Coulomb interaction strengths, $V_{i,j}$, with values given in Supplementary Tables 5, 6. With no intersite Coulomb interactions, a single conductance peak is observed (Extended Data Fig. 3a), whereas the inclusion of intersite Coulomb interactions gives rise to two conductance peaks, in which the separation between the peaks increases with increasing $V_{i,j}$. In addition to the presence of intersite Coulomb interactions, a small splitting (about 0.04 meV) in the topological states would arise owing to the non-zero $v/w$ ratio and the finite length of the chain. Here the topological states, exponentially localized at opposite ends of the chain, have a finite overlap. This small splitting, as presented in Extended Data Fig. 3a, is, however, too small to be observed experimentally.

## Data availability

The data pertaining to this study are available from the corresponding author upon reasonable request.

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

**Acknowledgements** The research was supported by Silicon Quantum Computing Pty Ltd and the Australian Research Council Centre of Excellence for Quantum Computation and Communication Technology (project number CE170100012). M.Y.S. acknowledges an Australian Research Council Laureate Fellowship.

**Author contributions** S.K.G., Y.H. and M.Y.S. conceived the project. M.K., H.G., M.B.D., J.G.K. and Y.C. fabricated the devices. M.K. and S.K.G. performed the measurements, analysed the data and performed the theoretical calculations. The manuscript was written by M.K., S.K.G., J.G.K. and M.Y.S. with input from all authors. M.Y.S supervised the project.

**Competing interests** The authors declare no competing interests.

**Additional information**
**Correspondence and requests for materials** should be addressed to M. Y. Simmons.

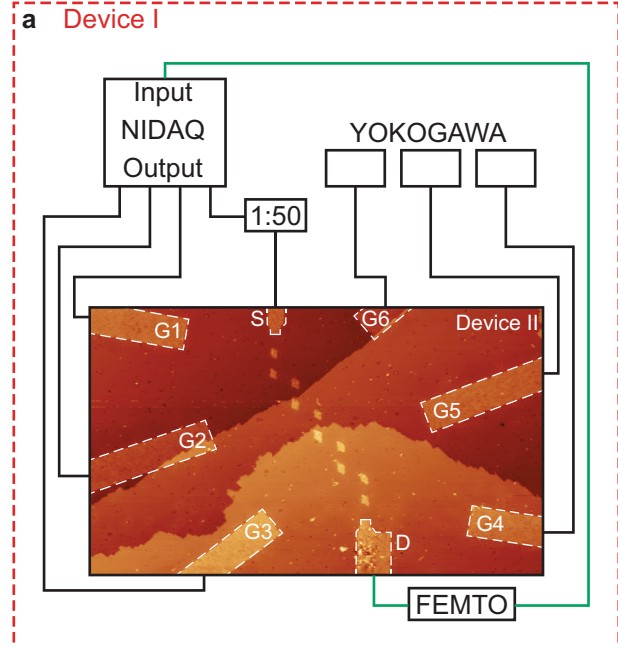

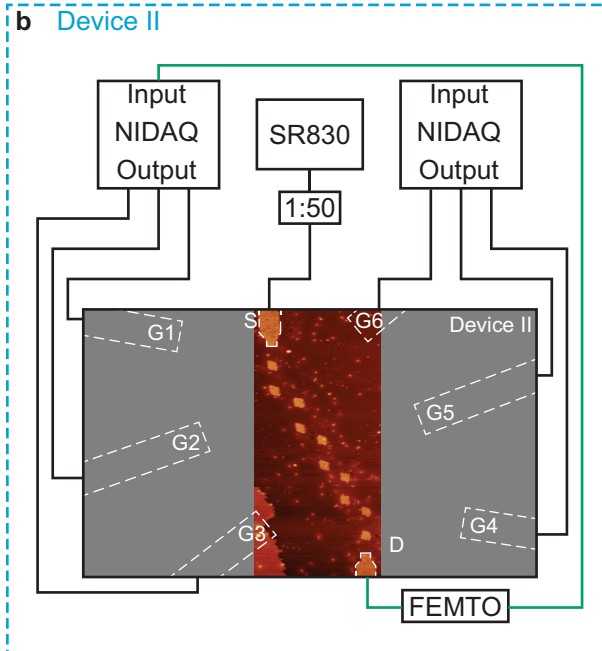

**Extended Data Fig. 1 | Experimental measurement setup. a**, Schematic of the experimental set up for Device I, showing all electrical connections to the device. A STM image of the device I is shown. The gate voltages are controlled by a NIDAQ and three Yokogawas, and the source voltage by the NIDAQ. The drain current (green line) is amplified via a FEMTO low noise amplifier and acquired by the NIDAQ. **b**, Schematic of the experimental setup for Device II, showing all electrical connections to the device. A STM image of the 10 dot array is shown, taken before patterning of the control gates. The gate voltages are controlled by two NIDAQs, and the source voltage controlled by a SR830 DSP lock-in amplifier. The drain current (green line) is amplified via a FEMTO low noise amplifier and acquired by the NIDAQ.

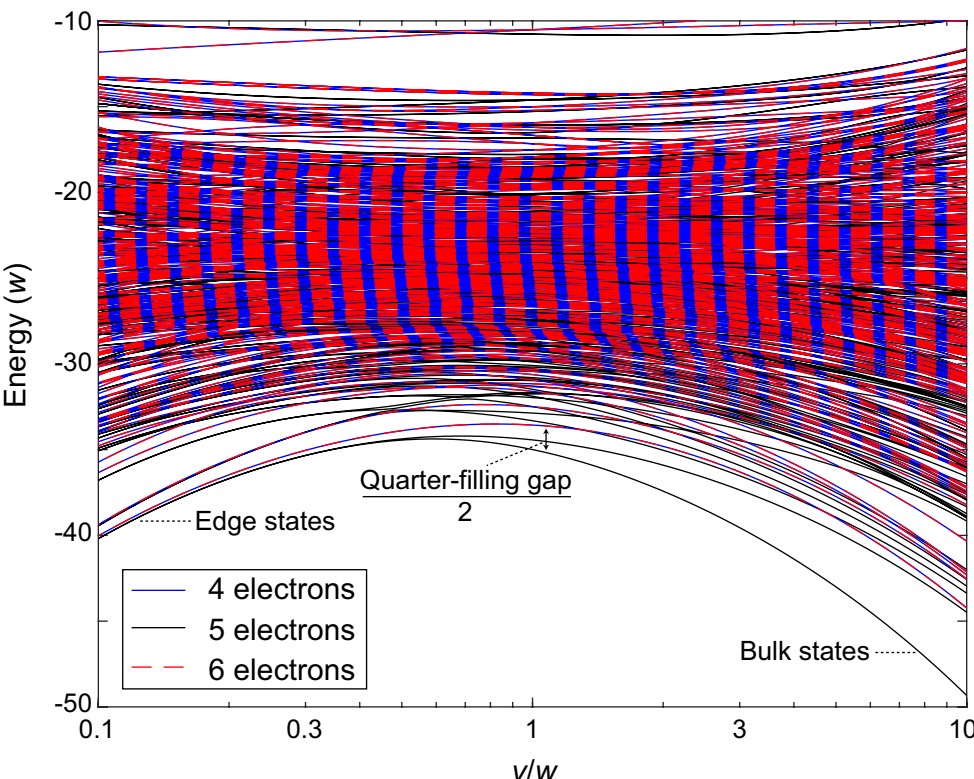

**Extended Data Fig. 2 | Energies of the many-body states in the interacting SSH model around quarter-filling.** The energies of $m+4$, $m+5$, and $m+6$ electron states for an array of 10 sites as a function of the tunnel coupling ratio, $v/w$. The energies are calculated at quarter-filling (that is, the ground state is always $m+5$ electrons). The ground state evolves from a singly degenerate state for $v/w>1$ (bulk-like states) to a two-fold degenerate $m+5$ electron state for $v/w<1$ (edge-like states). The quarter-filling energy gap between the $m+4$ and $m+6$ electron ground states reduces as $v/w<1$ resulting in nearly degenerate states, reflecting the almost zero-energy cost associated with adding an electron to the topological phase of the array, with a small energy gap due to the inter-site Coulomb interactions.

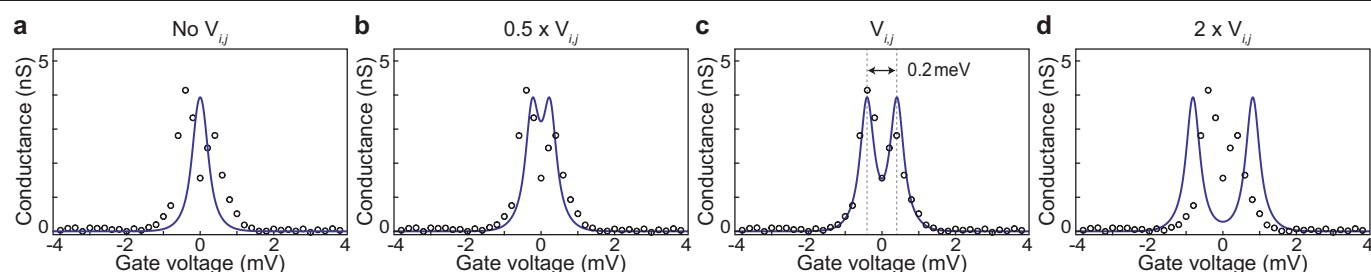

**Extended Data Fig. 3 | Splitting of the topological phase due to the presence of inter-site Coulomb interactions.** A comparison of the theoretically calculated conductance traces (blue lines) with the experimental data observed (circles) as the inter-site Coulomb interaction strength is varied from, **a**, no Coulomb interactions, to **b**, $0.5 \times V_{i,j}$, **c**, $V_{i,j}$, and **d**, $2 \times V_{i,j}$. In the case of no Coulomb interactions only a single peak is observed in the theoretical conductance trace, which does not match the roughly 0.2 meV splitting observed in the experimental data. As the Coulomb interaction strength is increased two peaks evolve with the splitting between the peaks increasing, with the roughly 0.2 meV splitting matching the experimental data in **c**.