## [Peer Review File · Nature]

Manuscript Title: Engineering topological states in atom-based semiconductor quantum dots

Reviewer Comments & Author Rebuttals

Reviewer Reports on the Initial Version:

Referees' comments:

Referee #1 (Remarks to the Author):

In this work, a quantum simulator built with quantum dot arrays based on phosphorous dopants in Si has been studied. There are so far only very few examples (and very limited in size) of semiconductor based quantum simulators. These would have an enormous potential to study, e.g., the many complex phases that arise in correlated electron systems. Local interactions can be specially large in dopant based quantum dots due to the large binding energies making these systems especially useful for simulation strong correlations.

In this manuscript, two tailored samples with 10 dot arrays in a zigzag configuration have been shown to hold, respectively, the topological and trivial solutions of the Su-Schrieffer-Heeger model with onsite (Hubbard) interactions (SSH). By precisely locating the dots, the required relations between the hoppings in the bipartite lattice of the SSH model have been engineered and the two solutions can be distinguished by studying the resulting transport features. Compared to electrostatically defined quantum dots, the dopant based dot arrays require much fewer gates simplifying the architecture and tuning procedure of the device. On the other hand, this would possibly imply a smaller range of parameters available to play with (in fact, two separate devices have been needed in order to observe the 2 qualitatively different solutions) but the proof of concept is really impressive anyway. I want to stress that the technological challenge for the realization of this work is enormous: it has been only possible after many years of accumulated expertise in the group.

The experimental work is described carefully in the manuscript plus supplementary material with many details about fabrication, the tuning of the gate potentials, the equipment used and the conditions for the measurements, etc. The particular choice of the dot size and the number of dopants they include is well justified in terms of guaranteeing the required uniformity of interaction potentials and getting to the sweet spot for the hopping values. There is also a thorough analysis of the effect of disorder.

The two devices give qualitatively different conductance which are ascribed to trivial and non-trivial (topological) states. The two states are well characterized and theoretical calculations are included in order to interpret the experimental data.

I find this work quite impressive due both to the engineering challenge that has been accomplished and to the significance of the experimental demonstration of the SSH interacting model. The results

are well founded and previous works are well referenced, as far as I am aware. I would certainly recommend publication of this manuscript after the following issues are solved:

1. It is not clear how the interacting model has been solved. Some explicit information in the supplementary material is needed.
2. An explanation to why quarter filling is special in this system is also missing. It could maybe help to have a figure like 1b for the interacting SSH model rather than the non-interacting one.
3. At the end of page 8 in the supplementary material, it is stated that the hoppings in different directions are different by a factor of 2 for the same distance. This dependence is well known as it is acknowledged in the manuscript. It would be useful to know the crystallographic directions with respect to the position of the dots. I haven't found that info in the manuscript or supplementary material.

Referee #2 (Remarks to the Author):

Using phosphorus donors in silicon, two devices containing ten quantum dot chains are implemented, with different separations between donors used to set tunnel coupling between dots. The devices implement the SSH model, with one device designed to be in the trivial regime and one to be in the topological regime. After careful optimization to find the energy at which transport can occur across the chain, the gate voltage is swept and the conductance peaks corresponding to electrons moving across the chain are measured. The measurements of the device in the trivial regime are used to simulate the results for the device in the topological regime. The use of quantum dots allows increased control of coupling between elements on the system than other platforms in which the SSH model has been studied. Compared to gate-defined quantum dots, far larger donor systems can be created, and the control over the various dot parameters is superior, in particular the constancy of the dot energies and tunnel couplings as nearby gates are swept. This is an impressive achievement, and indicates that donors may develop into a powerful platform for quantum simulation. I believe this is worthy of being published in Nature.

A few comments:

1) While much of the paper is well-written, the use of commas is quite poor and makes it significantly more challenging to understand. I recommend reading through it carefully and correcting them. For instance, these are the errors I fixed in the first paragraph:

Line 62: v and w, as shown

Line 68: range, which

Line 70: sites, while

Line 76: model been

Line 79: strength, however,

2) I found that the explanation of Figure 3d is too brief. The y-axis should be better defined; I assumed that it is the value of VSD at which transport turns on (i.e. the edge of the Stability map in

2b), but I'm not sure.

3) On line 243, it is indicated that tunneling between next-nearest neighbors is close to zero due to the staggered geometry. Given that the more typical configuration of a 1d-lattice of quantum dots is linear, this seems misleading. This is clarified in the supplement, where it is explained that the staggered geometry is optimal in terms of limiting tunnel coupling while maximizing the differential lever arm to the dots. It would be helpful if it were similarly discussed in the main manuscript.

4) On line 350, the voltage is referred to as being adjusted from N to $N+10$. It is more appropriate to say that the voltage is adjusted to tune the electron number from N to $N+10$. In the following line, it would be better to say that 7.9 is approximately twice 3.85.

5) I find the discussion of filling in the manuscript confusing. In the caption to Fig. 1e, N is defined to be the number of quantum dots. However, my reading of N as regards filling in the latter part of the paper is that it represents the electron number when the gate voltage is set to 0? If so, it would be appropriate to not use N for both (it is also used as "Iteration N " in Figure 2b, but there the context is clear). It would also be helpful to define N as regards filling.

6) Some discussion of disorder should be incorporated into the main manuscript. This is essential to the credibility of the report, and is covered comprehensively in the supplement.

7) At this early stage of quantum simulation with donors, it is acceptable that only transport measurements be used, but the results would be more convincing with localized measurements using charge sensors that showed the occupation of each of the quantum dots, and allowed for more direct extraction of the tunnel couplings of each dot. Currently, the paper rests on the combination of modeling the dots with transport measurements.

Referee #3 (Remarks to the Author):

In the following, I will provide my review following the recommended structure.

A. Summary of the key results:

The authors fabricated and characterized two SSH chains with electrostatic gates, via STM lithography and through-transport measurements. Their setup allows the authors to control the electron occupation in the artificial lattices - and thereby the electron-electron interaction strength.

B. Originality and significance: if not novel, please include reference.

Electron correlations are responsible for some of the most intriguing phenomena in condensed matter physics. They are also notoriously difficult to treat theoretically. An experimental platform that allows tuning the electron-electron interaction strength is therefore highly desirable. The present system in principle provides exactly such a platform. As such, I recommend publishing the present manuscript in Nature.

C. Data & methodology: validity of approach, quality of data, quality of presentation.

The experimental work represents a real tour-the-force and is very much state-of-the-art (as an experimentalist, it is more difficult for me to judge the theoretical effort). It naturally builds on previous work of the group (that has also developed most of the applied methodology). The validity of the applied approach (STM lithography to create well-defined nanometer scale devices and transport measurements on these devices) has been demonstrated independently in previous publications. In my view, the manuscript is beautifully written and clear.

D. Appropriate use of statistics and treatment of uncertainties.

If I am not mistaken, only two devices were fabricated (one based on the trivial SSH chain, and one with a topologically non-trivial chain). Given the efforts involved in device fabrication, this is acceptable. Other uncertainties are treated adequately.

E. Conclusions: robustness, validity, reliability

I would like the authors to comment on an issue related to the finite size of the chain, in combination with a non-zero w/v ratio. The topological states of the non-trivial SSH chain are only exponentially localized on the sites at the end of the chain. For a ten-site chain, and a $v/w = 0.265$ ratio, the states on opposite ends of the chain have a finite overlap. Consequently, these states (slightly) move away from zero energy. Could the ~ 0.2 meV splitting observed in Fig. 3f be the result of this split? In other words: at present, I am not convinced that this feature in Fig. 3f is a unique feature of the many-body SSH model. This is a rather important issue, as it is central to the use of this platform as a quantum simulator.

F. Suggested improvements: experiments, data for possible revision.

Very minor point: On page 4, right column, 'At zero source drain bias (red dashed line ...' The dashed line in Fig 3b is white, not red.

G. References: appropriate credit to previous work?

The main text includes appropriate references (both number and works). It might be useful to include some more references in the supplementary information to earlier work describing the device fabrication.

H. Clarity and context: lucidity of abstract/summary, appropriateness of abstract, introduction and conclusions

The manuscript is well-written and should be accessible for the non-specialist as well.

Author Rebuttals to Initial Comments:

Response to referee #1

We thank the referee for their comments and recommending our paper for publication in *Nature*. The referee lists a few comments which we address below (referee comments are in *blue*, our responses are in black, and excerpts/changes in the text are shown in *orange*):

1. *It is not clear how the interacting model has been solved. Some explicit information in the supplementary material is needed.*

We thank the referee for this comment. To solve the interacting many body SSH model we use an open-source python package, QmeQ. Into this model we describe the quantum dot Hamiltonian by the single particle states, $\sum \epsilon_i n_i + \sum t_{i,i+1} (c_i^\dagger c_{i+1} + h.c.)$, and the Coulomb matrix elements, $\sum V_{i,j} n_i n_j$, where the input values for the tunnel couplings are given in supplementary Table S6, and the inter-site Coulomb interaction strengths are given in supplementary Table S4 and S5. The Hamiltonian is constructed in the Fock basis, for example $|0010101110\rangle$ where a 1 indicates an electron at that quantum dot and 0 indicates the quantum dot is unoccupied, and diagonalised exactly to obtain the many-body eigenstates $|a\rangle$, $H = \sum E_a |a\rangle\langle a|$. Transport current through the quantum dot array is then calculated numerically using the Pauli master equation. The array is assumed to be weakly tunnel coupled to the source and drain leads at a temperature, T , with a density of states that follows a Fermi-distribution, $f(E) = [e^{(E-\mu)/T} + 1]^{-1}$. To clarify this, we have added the following description to the methods:

“The quantum dot array can be described by the extended Hubbard model with the Hamiltonian given by Equation 1. To theoretically solve the extended Hubbard model and calculate the parameters of the array the data was fit using an open-source python package Qmeq [45]. The quantum dot Hamiltonian is described by the single particle states, $\sum \epsilon_i n_i + \sum t_{i,i+1} (c_i^\dagger c_{i+1} + h.c.)$, and the Coulomb matrix elements, $\sum V_{i,j} n_i n_j$. The Hamiltonian is constructed in the Fock basis [45, 46], for example $|0010101110\rangle$ where a 1 indicates an electron at that quantum dot and 0 indicates the quantum dot is unoccupied, and diagonalised exactly to obtain the many-body eigenstates $|a\rangle$, $H = \sum E_a |a\rangle\langle a|$. Transport current through the quantum dot array is then calculated numerically using the Pauli master equation. The array is assumed to be weakly coupled to the source and drain leads at a temperature, T , with a density of states that follows a Fermi-distribution $f(E)$.”

2. *An explanation to why quarter filling is special in this system is also missing. It could maybe help to have a figure like 1b for the interacting SSH model rather than the non-interacting one.*

We agree with the referee that an explanation of why quarter filling is special in this system would be helpful. The relevance of quarter-filling becomes significant when we consider the electron occupation of each of the quantum dot “dimers”. For an array of 10 quantum dots in which each dot can host two electrons, quarter-filling corresponds to where there are 5 electrons shared across the dots. Here, if the array is in the trivial phase, the electrons will simply arrange themselves into each dimer such that they are spread evenly across the array. In contrast, if the array is in the topological configuration at quarter-filling, 4 electrons arrange themselves in the middle of the array (similar to the trivial phase). However, the 5th electron cannot occupy a dimer since the last two empty quantum dots are at each end of the array. It is this topology that forces the 5th electron to be in a superposition of the two end quantum dots leading to the interesting topological/conductance signatures observed in the experiment. This scenario only occurs at quarter-filling for electrons, hence the experiment was focused around quarter-filling of the array. We have expanded our explanation to cover this briefly in the main text and then added an extra figure in the supplementary as requested:

“In Fig. 1c,d we show the 1024 calculated multi-electron energy levels based on the two experimental arrays fabricated in the trivial, and topological phases respectively, order by the number of electrons in the quantum dot array at quarter-filling. For arrays of 10 quantum dots in which each dot can host two electrons, quarter-filling corresponds to where there are 5 electrons shared across the dots. Here, if the array is in the trivial phase, the electrons will simply arrange themselves into each dimer such that they are spread evenly across the array. In contrast, if the array is in the topological configuration at quarter-filling, 4 electrons arrange themselves in the middle of the array (similar to the trivial phase). However, the 5th electron cannot occupy a dimer. We focus on quarter-filling of the array since the interacting topological states involve this 5th electron becoming localised to each end of the array.”

The purpose of Figure 1b was to introduce the concept of the topological phases of the SSH model, which is easily conveyed using the non-interacting model. An energy level diagram of the interacting topological states contains 1024 many-body states of the array as a function of v/w , and as such is more complex. However, we agree this is also useful in the paper, and as requested we have added this to section V of the supplementary to show how the ground states at quarter-filling vary as a function of v/w . We have copied this new figure on the next page and added the following text to the Methods material:

“Analogous to the non-interacting energy-level diagram as shown in Fig. 1b of the main text we also calculated the energy-level diagram of the interacting system. As with Fig. 3a in the main text we assume the inter-site Coulomb terms follow a $d^{-1.5}$ dependence [47] and that the electrochemical potentials of the quantum dots are tuned via $\epsilon_i = -\sum V_{ij}$. Extended Data Figure 2 shows the energy spectrum for an array of 10 sites for the $m + 4$, $m + 5$, and $m + 6$ electron states at quarter-filling showing the more complex excited state spectrum of the many-body states of the array.

For $v/w > 1$, at quarter-filling ($m + 5$ electrons), there is a large gap between the ground $m + 5$ state and the lowest $m + 4$ and $m + 6$ electron states with a single ground state (labelled ‘bulk states’ analogous to Fig. 1b of the main text). By changing the tunnel couplings such that $v/w < 1$ the ground state becomes doubly-degenerate with $m + 5$ electrons (labelled ‘edge states’ analogous to Fig. 1b of the main text). Here the quarter-filling gap is greatly reduced resulting in nearly degenerate states with states of differing electron numbers. The small energy gap at $v/w < 1$, is observed in the conductance trace in Fig. 3f of the main text and reflects the inter-site Coulomb interactions present in the system.”

Extended Data Figure 2: Energies of the many body states in the interacting SSH model around quarter-filling. The energies of $m + 4$, $m + 5$, and $m + 6$ electron states for an array of 10 sites as a function of the tunnel coupling ratio, v/w . The energies are calculated at quarter-filling (that is, the ground state is always $m + 5$ electrons). The ground state evolves from a singly degenerate state for $v/w > 1$ (bulk-like states) to a two-fold degenerate $m + 5$ electron state for $v/w < 1$ (edge-like states). The quarter-filling energy gap between the $m + 4$ and $m + 6$ electron ground states reduces as $v/w < 1$ resulting in nearly degenerate states, reflecting the almost zero-energy cost associated with adding an electron to the topological phase of the array, with a small energy gap due to the inter-site Coulomb interactions.

In plotting the energies of the interacting SSH model around quarter-filling for this new figure we realised that the conductance curve in Figure 3a of the original manuscript (see image “old version” on the next page) should have been symmetric about the zero in the electrochemical potential indicating that the quantum dot array is perfectly aligned. We discovered an error in the plotting code used to generate the figure, where the V_{ij} terms as a function of the inter-dot separation had not been updated between each line of the colour plot for different values of v/w . We have corrected this in the “new version” below. The updated version of Fig. 3a has no impact on any of the outcomes of paper since all the theoretical fits in the other subfigures of Fig. 3 used the V_{ij} terms directly calculated from the electrostatic modelling.

To clarify the importance of the symmetric energy alignment we have plotted this new version of the conductance as a function of energy in units of w instead of electrochemical potential which assumes a certain lever arm (~ 0.3). This both emphasises that this is a theoretical calculation and shows the symmetric transition from the trivial to topological state, as expected. Both figures show the transition from a trivial configuration with 10 peaks at $v/w > 1$ to a topological phase at $v/w < 1$ with two closely separated peaks.

3. *At the end of page 8 in the supplementary material, it is stated that the hoppings in different directions are different by a factor of 2 for the same distance. This dependence is well known as it is acknowledged in the manuscript. It would be useful to know the crystallographic directions with respect to the position of the dots. I haven't found that info in the manuscript or supplementary material.*

We have added the crystallographic directions to the methods section of the paper:

“... crystallographic orientation of the quantum dots with v ($\sim \langle 100 \rangle$) and w ($\sim \langle 120 \rangle$) having different scaling factors ...”

Response to referee #2

We thank the referee for their time in reviewing our manuscript. We have addressed their comments and questions below:

1. *While much of the paper is well-written, the use of commas is quite poor and makes it significantly more challenging to understand. I recommend reading through it carefully and correcting them. For instance, these are the errors I fixed in the first paragraph:*

Line 62: v and w, as shown

Line 68: range, which

Line 70: sites, while

Line 76: model been

Line 79: strength, however,

We agree with the referee's comments and have revised the text to make corrections where necessary.

2. *I found that the explanation of Figure 3d is too brief. The y-axis should be better defined; I assumed that it is the value of VSD at which transport turns on (i.e. the edge of the Stability map in 2b), but I'm not sure.*

We thank the referee for picking this up. In the previous version of the manuscript, Fig. 3d had the y-axis labelled as 'Voltage (meV)'. However, this was a typo and the y-axis should have read 'Energy (meV)'. We have fixed this error in the current version of the manuscript. We have also added another sentence in the main text further explaining the origin of the figure:

"Figure 3d shows the width of the experimentally measured stability regions, obtained by determining the energy between adjacent conductance peaks, compared to theoretical calculations based on electrostatic modelling with a tunnel coupling ratio $v/w = 2.08$."

3. *On line 243, it is indicated that tunneling between next-nearest neighbors is close to zero due to the staggered geometry. Given that the more typical configuration of a 1d-lattice of quantum dots is linear, this seems misleading. This is clarified in the supplement, where it is explained that the staggered geometry is optimal in terms of limiting tunnel coupling while maximizing the differential lever arm to the dots. It would be helpful if it were similarly discussed in the main manuscript.*

Since we don't wish to cause any confusion, we have clarified in the main text that the use of the staggered geometry ensures that non-nearest neighbour tunnelling is suppressed while the differential lever-arms are maximised:

"By staggering the quantum dot array we have ensured that the non-nearest neighbour tunnelling is exponentially suppressed with an estimated $t_{i,i+2}/t_{i,i+1} \approx 0.01$, ensuring electron transport occurs in series through the array, while maximising the differential lever-arms to the dots (see Supplementary I for a complete description of the device design)."

4. *On line 350, the voltage is referred to as being adjusted from N to N+10. It is more appropriate to say that the voltage is adjusted to tune the electron number from N to N+10. In the following line, it would be better to say that 7.9 is approximately twice 3.85.*

We apologise for the confusing wording and have adjusted the paper according to the referee's suggestions:

"We control the electron-filling of the quantum dot array by adjusting the gate voltages to tune the electron number from m to $m + 10$ (half-filling)."

"... (7.9 meV which is approximately twice the gap of 3.85 meV in Fig 1c.) ..."

5. *I find the discussion of filling in the manuscript confusing. In the caption to Fig. 1e, N is defined to be the number of quantum dots. However, my reading of N as regards filling in the latter part of the paper is that it represents the electron number when the gate voltage is set to 0? If so, it would be appropriate to not use N for both (it is also used as "Iteration N" in Figure 2b, but there the context is clear). It would also be helpful to define N as regards filling.*

The referee raises an excellent point that we have used the notation N to represent the number of quantum dots, the number of electrons, and the number of iterations in the alignment procedure. We also note that we have used i with regards to both the quantum dot number and the electron number. This is confusing and we

have rectified it. We have changed the terminology to refer to N quantum dots, m and k electrons and dropped the index for the iteration procedure throughout the manuscript. We clarify that the number of electrons is taken as $m + k$ since the quantum dots are not fully depleted at zero gate bias and as such there is a number (m) of electrons already present on the dot. This new definition helps to clarify the difference and does not change any of the results of the manuscript.

6. *Some discussion of disorder should be incorporated into the main manuscript. This is essential to the credibility of the report, and is covered comprehensively in the supplement.*

We have moved part of the discussion on the effects of disorder on the experimental results from the supplementary material to the main text.

“... small misalignments of the quantum dot electrochemical potentials, which gives rise to on-site disorder, that causes small shifts in the conductance peaks such that the peak structure is no longer symmetric around zero (see supplementary III).”

7. *At this early stage of quantum simulation with donors, it is acceptable that only transport measurements be used, but the results would be more convincing with localized measurements using charge sensors that showed the occupation of each of the quantum dots, and allowed for more direct extraction of the tunnel couplings of each dot. Currently, the paper rests on the combination of modeling the dots with transport measurements.*

We agree – future experiments will indeed examine the quantum dot occupation via charge sensing. The current experimental results offer a proof-of-principle demonstration of quantum simulation using donor-based quantum dots, and the technology developed now opens the door to many future experiments in the development of this platform for quantum simulation beyond the classical computing limit. We have added this to the future directions at the end of the paper:

“Future work will focus on extending the size of the arrays, the incorporation of charge sensors, and extending the simulations to engineered 2D lattices.”

Response to referee #3

We thank referee #3 for their comments. They raised a couple of minor clarifications and an important question regarding the observed splitting of the conductance peaks in the topological device. We address their question below.

1. *I would like the authors to comment on an issue related to the finite size of the chain, in combination with a non-zero v/w ratio. The topological states of the non-trivial SSH chain are only exponentially localized on the sites at the end of the chain. For a ten-site chain, and a $v/w = 0.265$ ratio, the states on opposite ends of the chain have a finite overlap. Consequently, these states (slightly) move away from zero energy. Could the ~ 0.2 meV splitting observed in Fig. 3f be the result of this split? In other words: at present, I am not convinced that this feature in Fig. 3f is a unique feature of the many-body SSH model. This is a rather important issue, as it is central to the use of this platform as a quantum simulator.*

The referee queries whether the origin of the approximately 0.2 meV splitting of the topological states observed in Fig. 3f can arise from the finite size chain, with a non-zero v/w ratio, in which the states at the opposite ends of the chain will still have a finite overlap causing a shift in the energy states slightly away from zero energy. We have calculated the expected splitting from the finite overlap that occurs for the devices fabricated with no Coulomb interaction terms and found it to be only ~ 0.04 meV. For the non-interacting case therefore, we would not be able to resolve this splitting experimentally and only a single conductance peak would be observed. It is only when we consider the interacting many-body SSH model that we get the approximately 0.2 meV splitting observed in the experiment. We have included an extra figure in section V of the Methods to show how the interaction strength changes the splitting observed:

“The presence of inter-site Coulomb interactions gives rise to a ~ 0.2 meV splitting observed in the topological states as seen in Fig. 3f of the main text. Extended Data Figure 3 shows the calculated conductance for the topological phase for varying inter-site Coulomb interaction strengths, V_{ij} , with values given in Table S5 and Table S6 of the supplementary. With no inter-site Coulomb interactions a single conductance peak is observed (Extended Data Fig. 3a), whereas with the inclusion of the inter-site Coulomb interactions gives rise to two conductance peaks, in which the separation between the peaks increases with increasing V_{ij} . In addition to the presence of inter-site Coulomb interactions, a small splitting (~ 0.04 meV) in the topological states would arise due to the non-zero v/w ratio and finite length of the chain. Here the topological states, exponentially localised at opposite ends of the chain, have a finite overlap. This small splitting, as presented in Extended Data Fig. 3a, is however too small to be observed experimentally.”

Extended Data Figure 3. **Splitting of topological phase due to the presence of inter-site Coulomb interactions.** A comparison of the theoretically calculated conductance traces (blue lines) with the experimental data observed (circles) as the inter-site Coulomb interaction strength is varied from, **a**, no Coulomb interactions, to **b**, $0.5 \times V_{ij}$, **c**, V_{ij} , and **d**, $2 \times V_{ij}$. In the case of no Coulomb interactions only a single peak is observed in the theoretical conductance trace, which doesn't match the ~ 0.2 meV splitting observed in the experimental data. As the Coulomb interaction strength is increased two peaks evolve with the splitting between the peaks increasing, with the ~ 0.2 meV splitting matching the experimental data in **c**.

2. *Very minor point: On page 4, right column, 'At zero source drain bias (red dashed line ...' The dashed line in Fig 3b is white, not red.*

We thank the referee for picking up this typo. We have fixed the text to say “white line”.

“... (dashed white line ...”

3. *The main text includes appropriate references (both number and works). It might be useful to include some more references in the supplementary information to earlier work describing the device fabrication.*

We have added the following additional references for completeness, along with the accompanying line to the methods section:

“A more detailed description of scanning tunnelling microscope hydrogen lithography and the further device processing to electrically contact the device can be found in references [42-44]”

[42] F. J. Ruess, W. Pok, T. C. G. Reusch, M. J. Butcher, K. E. J. Goh, L. Oberbeck, G. Scappucci, A. R. Hamilton, M. Y. Simmons, Realization of atomically controlled dopant devices in silicon, *Small*, **3**, 563 (2007)

[43] M. A. Broome, S. K. Gorman, M. G. House, S. J. Hile, J. G. Keizer, D. Keith, C. D. Hill, T. F. Watson, W. J. Baker, L. C. L. Hollenberg, and M. Y. Simmons. Two-electron spin correlations in precision placed donors in silicon. *Nature Communications*, 9(1), mar 2018.

[44] L. Fricke, S.J. Hiles, L. Kranz, Y. Chung, Y. He, P. Pakkiam, J.G. Keizer, M.G. House and M.Y. Simmons. Coherent control of a donor-molecule electron spin qubit in silicon. *Nature Communications* 12, 3323 (2020).

Minor manuscript correction:

We have detected a small typographical error in Table 1 where we inadvertently quoted the maximum value of the tunnel coupling (5meV) and not the mean tunnel coupling (3.4meV) determined in the experiment from the average of all tunnel couplings stated in supplementary Table S6.

We have corrected this value in Table 1 and note that as a typographical error it does not have any impact on the qualitative or quantitative results of the experimental/theoretical results.